# NiemaGraphGen: A memory-efficient global-scale contact network simulation toolkit

Niema Moshiri[1],*

1 Department of Computer Science & Engineering, UC San Diego, La Jolla, 92093, USA

## ABSTRACT

Epidemic simulations require the ability to sample contact networks from various random graph models. Existing methods can simulate city-scale or even country-scale contact networks, but they are unable to feasibly simulate global-scale contact networks due to high memory consumption. NiemaGraphGen (NGG) is a memory-efficient graph generation tool that enables the simulation of global-scale contact networks. NGG avoids storing the entire graph in memory and is instead intended to be used in a data streaming pipeline, resulting in memory consumption that is orders of magnitude smaller than existing tools. NGG provides a massively-scalable solution for simulating social contact networks, enabling global-scale epidemic simulation studies.

**Subjects** Software and Workflows, Bioinformatics, Statistics and Probability

## STATEMENT OF NEED

The ability to simulate epidemics enables the evaluation of the effectiveness of molecular epidemiological tools [1] as well as the inference of critical public health information, such as the time of zoonosis of SARS-CoV-2 [2]. Epidemic simulation frameworks such as FAVITES simulate a random contact network, a random transmission network spread along the contact network, a viral phylogeny constrained by the transmission network, and a random viral sequence evolutionary process (e.g. single gene/protein, whole genome) along the phylogeny [3]. The spread of viral pathogens is driven by social contact networks [4], and the structure of the underlying contact network across which a virus transmits is heavily influenced by the mode of disease transmission, necessitating a proper match between pathogen and network model when designing epidemic simulation experiments [5]. As a result, epidemic simulation frameworks such as FAVITES require the flexibility to simulate contact networks under a wide selection of network models.

The two most popular existing tools for simulating networks under various stochastic models are NetworkX [6], which is available as a Python package, and iGraph [7], which is available as a C library with R, Python, and Mathematica interfaces. While these tools support random network generation, they place an emphasis on the analysis and manipulation of networks, and as a result, they require loading the entire network in memory. This is feasible for community-level or even city-scale epidemic simulations, but when simulating global-scale pandemics such as COVID-19, the memory consumption becomes prohibitively large.

Tools such as EpiSims [8], EpiSimdemics [9], and EpiFast [10] provide efficient solutions for simulating transmission networks *along* a *given* contact network, but they do not

**Submitted:** 28 November 2021

\* E-mail: niema@ucsd.edu

Preprint submitted at https://arxiv.org/abs/2201.04625

simulate the contact network itself. For simulating contact networks, cuPPA [11] and cuPPA-Hash [12] provide GPU-accelerated solutions for massively-parallelized simulation of ultra-large scale-free networks under the Copy Model [13], but they do not support the simulation of contact networks under other graph models. Which is a critical feature for epidemiologists hoping to fine-tune simulations to the contact patterns of a given outbreak or population of interest.

## IMPLEMENTATION

NiemaGraphGen (NGG) is a memory-efficient undirected graph generation tool that enables the simulation of global-scale contact networks. NGG is intended to be used in data-streaming epidemic simulation pipelines and thus avoids storing the entire contact network in memory, resulting in faster runtime as well as memory consumption that is orders of magnitude smaller than existing tools (Figure 1).

NGG is written in C++ and has no dependencies beyond a modern C++ compiler (and optionally the command line `make` tool for convenience). When NGG is compiled, a separate executable is produced for each model. NGG is also available via a Docker container on DockerHub (niemasd/niemagraphgen). NGG currently supports the following stochastic and deterministic models: Barabási–Albert [14], Barbell, Complete, Cycle, Empty, Erdős–Rényi [15], Newman–Watts–Strogatz [16], Path, and Ring Lattice.

By default, NGG uses 4-byte unsigned integers to represent nodes in the network, which supports networks with up to $2^{32} - 1 \approx 4.3$ billion nodes, but users can use 2-byte (up to $2^{16} - 1 = 65,535$ nodes) or 1-byte (up to $2^8 - 1 = 255$ nodes) unsigned integers to reduce memory consumption, or they can use 8-byte unsigned integers (up to $2^{64} - 1 \approx 18$ quintillion nodes) to support larger networks at the cost of higher memory consumption.

By default, NGG outputs networks in the tab-delimited edge list format used by FAVITES [3]. Output files in this format can then be used as input files within FAVITES, which will then be able to simulate a transmission network, viral phylogeny, and sequences along the given contact network. However, for ultra-large simulation studies, plain-text edge list representations of networks may result in extremely large files. To address this NGG also implements a proprietary compact binary output format that uses exactly $2b|E| + 1$ bytes to represent a network with $|E|$ edges in which nodes are represented using $b$-byte unsigned integers. Both supported output formats are highly structured and can thus be compressed reasonably well using standard compression tools (e.g. `gzip`). FAVITES does not currently support this compact binary format, so contact networks output in this binary format will not be usable as input files in the current version of FAVITES (v1.2.8), but support for this binary format will be implemented into FAVITES in the near future. Code examples for loading contact networks in NGG's output formats can be found in the NGG GitHub Wiki (https://github.com/niemasd/NiemaGraphGen/wiki).

## Memory-efficient graph sampling

In this subsection, we discuss the memory-efficient graph sampling algorithms implemented within NGG. Most models implemented in NGG are sampled in $\mathcal{O}(1)$ memory.

### *Complete graph*

The complete graph, in which every node has an edge to every other node, is trivial to sample in $\mathcal{O}(1)$ memory (Algorithm 1).



---

**Algorithm 1** Sample complete graph

---

**procedure** COMPLETE($n$)
 **for** $u \leftarrow 0$ to $n - 2$ **do**
 **for** $v \leftarrow u + 1$ to $n - 1$ **do**
 Output edge $(u, v)$

---

### Path graph

The path graph, in which $n$ nodes are connected in a linear path, is trivial to sample in $\mathcal{O}(1)$ memory (Algorithm 2).

---

**Algorithm 2** Sample path graph

---

**procedure** PATH($n$)
 **for** $u \leftarrow 0$ to $n - 2$ **do**
 Output edge $(u, u + 1)$

---

### Barbell graph

The barbell graph, which consists of two complete graphs with $n_1$ nodes (Algorithm 1) connected by a path graph with $n_2$ nodes (Algorithm 2), can be sampled in $\mathcal{O}(1)$ memory (Algorithm 3).

---

**Algorithm 3** Sample barbell graph

---

**procedure** BARBELL($n_1, n_2$)
 ▷ Sample *Complete* ($n_1$) twice
 **for** $u \leftarrow 0$ to $n_1 - 2$ **do**
 **for** $v \leftarrow u + 1$ to $n_1 - 1$ **do**
 Output edge $(u, v)$ ▷ *Complete* $(n_1)_1$
 Output edge $(n_1 + u, n_1 + v + u)$ ▷ *Complete* $(n_1)_2$
 ▷ Sample *Path* ($n_2$)
 **for** $u \leftarrow 2n_1$ to $2n_1 + n_2 - 2$ **do**
 Output edge $(u, u + 1)$
 ▷ Connect *Complete* $(n_1)_1$ and *Complete* $(n_1)_2$ to *Path* ($n_2$)
 Output edge $(0, 2n_1)$
 Output edge $(n_1, 2n_1 + n_2 - 1)$

---

### Cycle graph

The cycle graph, which consists of a single $n$-node cycle, is trivial to sample in $\mathcal{O}(1)$ memory: it is simply a path graph (Algorithm 2) with a single additional edge connecting the start and end nodes (Algorithm 4).

---

**Algorithm 4** Sample cycle graph

---

**procedure** CYCLE($n$)
 *Path* ($n$)
 Output edge $(0, n - 1)$

---



### Ring lattice graph

The ring lattice graph, in which every node has an edge to each of its *k* neighbors (where *k* must be even), is essentially a generalization of the cycle graph. Specifically, *Cycle* (*n*) is equivalent to *RingLattice* (*n*, 2). The ring lattice graph can be sampled in $\mathcal{O}(1)$ memory (Algorithm 5).

---

**Algorithm 5** Sample ring lattice graph

---
**procedure** RINGLATTICE(*n, k*)
 **for** $u \leftarrow 0$ to $n - 1$ **do**
 **for** $d \leftarrow 1$ to $k/2$ **do**
 Output edge $(u, (u + i) \bmod n)$

---

### Erdős–Rényi model

The Erdős–Rényi model is a random graph model for generating networks, and it has two parameters: the total number of nodes in the network (*n*) and the probability that any of the $\binom{n}{2}$ possible edges is included (*p*). A naive algorithm can be used to sample graphs under the model in $\mathcal{O}(1)$ memory (Algorithm 6).

---

**Algorithm 6** Sample Erdős–Rényi model (naive)

---
1: **procedure** ERDOSRENYINAIVE(*n, p*)
2: **for** $u \leftarrow 0$ to $n - 2$ **do**
3: **for** $v \leftarrow u + 1$ to $n - 1$ **do**
4: **if** *Bernoulli* (*p*) = 1 **then**
5: Output edge $(u, v)$

---

However, the time complexity of the naive algorithm is $\mathcal{O}(n^2)$, making it unsuitable for ultra-large large networks. Instead, an alternative algorithm can also be implemented in $\mathcal{O}(1)$ memory (Algorithm 7), which is faster than the naive algorithm when the expected number of edges $(p\binom{n}{2})$ is relatively low (i.e., the network is relatively sparse) [17], as is the case with social contact networks.

---

**Algorithm 7** Sample Erdős–Rényi model

---
1: **procedure** ERDOSRENYIFAST(*n, p*)
2: $u \leftarrow 1$
3: $v \leftarrow -1$
4: **while** $u < n$ **do**
5: $v \leftarrow v + 1 + \frac{\ln(1 - Uniform(0,1))}{\ln(1-p)}$
6: **while** $v \geq u$ and $u < n$ **do**
7: $v \leftarrow v - u$
8: $u \leftarrow u + 1$
9: **if** $u < n$ **then**
10: Output edge $(u, v)$

---

### Barabási–Albert model

The Barabási–Albert model is a random graph model for generating scale-free networks, and it has two parameters: the total number of nodes in the network (*n*) and the number of edges to attach from new nodes to existing nodes (*m*). An algorithm exists to sample graphs



under the model in $\mathcal{O}(nm)$ memory. Graphs sampled under *BarabasiAlbert* (*n*, *m*) will have exactly *m* (*n* − *m*) edges, with exactly *m* targets selected during each iteration of the sampling algorithm. Thus, when implementing the sampling algorithm, memory for *repeat* and *targets* can be reserved up-front to avoid array resizing operations during the algorithm (Algorithm 8).

---

**Algorithm 8** Sample Barabási−Albert model

1: **procedure** BARABASIALBERT(*n*, *m*)
2:     *repeat* ← *ArrayList* ({0, 1, . . . , *m* − 1})
3:     *targets* ← *HashTable* (*empty*)
4:     **for** *u* ← *m* to *n* − 1 **do**
5:         Clear *targets*
6:         **while** |*targets*| < *m* **do**
7:             Add *Uniform* (*repeat*) to *targets*
8:         **for all** *v* ∈ *targets* **do**
9:             Output edge (*u*, *v*)
10:            *repeat* ← *repeat* + {*u*, *v*}

---

### Newman–Watts–Strogatz model

The Newman–Watts–Strogatz model, an extension of the Watts–Strogatz model [18], is a random graph model for generating connected networks with small-world properties. Unlike the Watts–Strogatz model, which may yield in disconnected graphs, the Newman–Watts–Strogatz model is guaranteed to yield connected graphs. The Newman–Watts–Strogatz model begins by sampling *RingLattice* (*n*, *k*), and for each edge (*u*, *v*) in in the initial ring lattice, a new "shortcut" edge (*u*, *w*) is added with probability *p*. This motivates a naive sampling algorithm (Algorithm 9).

---

**Algorithm 9** Sample Newman−Watts−Strogatz model (naive)

1: **procedure** NEWMANWATTSSTROGATZNAIVE(*n*, *k*, *p*)
2:     *ring* ← *HashTable* (*RingLattice* (*n*, *k*))
3:     *shortcuts* ← *HashTable* (*empty*)
4:     **for all** (*u*, *v*) ∈ *ring* **do**
5:         **if** *Bernoulli* (*p*) = 1 **then**
6:             *w* ← *u*
7:             **while** *w* = *u* or (*u*, *w*) ∈ *ring* ∪ *shortcuts* **do**
8:                 *w* ← *Uniform* (0, *n* − 1)
9:             Output edge (*u*, *w*)
10:            Add edge (*u*, *w*) to *shortcuts*

---

However, the naive algorithm requires all edges of the graph to be stored in memory, which results in prohibitively large memory requirements for ultra-large networks. An alternative memory-efficient algorithm can be devised. There are *n* nodes, and in the original ring lattice, each node has *k* edges. Therefore, the initial ring lattice graph has $nk/2$ undirected edges, meaning we sample from *Bernoulli* (*p*) exactly $nk/2$. The total number of successful Bernoulli trials is thus a single sampling from *Binomial* ($nk/2$, *p*). Further, each node has *n* − *k* −1 possible new edges that can be added during the "shortcut"-adding step;

these edges can be represented by a matrix with $n$ rows (representing $u$) and $n - k - 1$ columns (representing $w$):

$$
\begin{array}{c}
0: \\
1: \\
2: \\
... \\
i: \\
...
\end{array}
\left[
\begin{array}{ccccc}
k/2 + 1 & k/2 + 2 & ... & (n - k/2 - 1)\bmod n \\
k/2 + 2 & k/2 + 3 & ... & (n - k/2 - 0)\bmod n \\
k/2 + 3 & k/2 + 4 & ... & (n - k/2 + 1)\bmod n \\
... & ... & ... & ... \\
k/2 + i & ... & ... & (n - k/2 + i - 1)\bmod n \\
... & ... & ... & ...
\end{array}
\right]
\tag{1}
$$

If $(u, v)$ is selected, then $(v, u)$ cannot be selected because the graph is undirected. Thus, we can disregard the bottom-right portion of the matrix. We can then represent each cell of the matrix with its corresponding index in an array representation. For example, for $n = 7$ and $k = 2$ ($X$ denotes "disregarded"):

$$
\begin{array}{c}
0: \\
1: \\
2: \\
3: \\
4: \\
5: \\
6:
\end{array}
\left[
\begin{array}{cccc}
2 & 3 & 4 & 5 \\
3 & 4 & 5 & 6 \\
4 & 5 & 6 & 0 \\
5 & 6 & 0 & 1 \\
6 & 0 & 1 & 2 \\
0 & 1 & 2 & 3 \\
1 & 2 & 3 & 4
\end{array}
\right]
\rightarrow
\begin{array}{c}
0: \\
1: \\
2: \\
3: \\
4: \\
5: \\
6:
\end{array}
\left[
\begin{array}{cccc}
2 & 3 & 4 & 5 \\
3 & 4 & 5 & 6 \\
4 & 5 & 6 & X \\
5 & 6 & X & X \\
6 & X & X & X \\
X & X & X & X \\
X & X & X & X
\end{array}
\right]
\rightarrow
\left[
\begin{array}{cccc}
0 & 1 & 2 & 3 \\
4 & 5 & 6 & 7 \\
8 & 9 & 10 & X \\
11 & 12 & X & X \\
13 & X & X & X \\
X & X & X & X \\
X & X & X & X
\end{array}
\right]
\tag{2}
$$

With this representation, sampling "shortcut" edges can be reduced to an efficient algorithm: randomly select a collection of $Binomial\,(nk/2, p)$ integers from $Uniform(0, \frac{n(n-k-1)}{2} - 1)$ without replacement, then map from the selected integers to their corresponding cells in the matrix, and finally map from cells in the matrix to edges $(u, w)$.

Define a "full" row to be a row without any $X$ symbols (i.e., no disregarded cells), and define an "empty" row to be a row that only contains $X$ symbols (i.e., all cells were disregarded). The last column in the first row contains node $n - k/2 - 1$, and the last column in the last full row has node $n - 1$, so there are $(n - 1) - (n - k/2 + 1) + 1 = k/2 + 1$ non-empty rows: 0 through $k/2$. Thus, for rows 0 through $k/2$ (i.e., the full rows of the matrix), we can imagine the following representation in which cells are filled with the corresponding index of the array representation of the matrix:

$$
\begin{array}{c}
0: \\
1: \\
2: \\
... \\
i: \\
... \\
k/2:
\end{array}
\left[
\begin{array}{cccc}
0 & 1 & ... & n - k - 2 \\
n - k - 1 & n - k & ... & 2(n - k - 1) - 1 \\
2(n - k - 1) & 2(n - k - 1) + 1 & ... & 3(n - k - 1) - 1 \\
... & ... & ... & ... \\
i(n - k - 1) & i(n - k - 1) + 1 & ... & (i + 1)(n - k - 1) - 1 \\
... & ... & ... & ... \\
\frac{k}{2}(n - k - 1) & \frac{k}{2}(n - k - 1) + 1 & ... & (\frac{k}{2} + 1)(n - k - 1) - 1
\end{array}
\right]
\tag{3}
$$

Row $k/2 + 1$ has exactly 1 empty cell, row $k/2 + 2$ has exactly 2 empty cells, etc. Thus, the first row that is completely empty (i.e., $n - k - 1$ empty cells) is row $k/2 + (n - k - 1) = n - k/2 - 1$. Thus, the remaining portion of the matrix from which



"shortcuts" can be sampled can be represented as follows ($X$ denotes "disregarded", and $Y$ denotes "not disregarded"):

$$
\begin{array}{r}
k/2 + 1: \\
k/2 + 2: \\
\ldots \\
n - k/2 - 2:
\end{array}
\begin{bmatrix}
Y & Y & Y & \ldots & Y & X \\
Y & Y & Y & \ldots & X & X \\
\ldots & \ldots & \ldots & \ldots & \ldots & \ldots \\
Y & X & X & \ldots & X & X
\end{bmatrix}
\tag{4}
$$

This is simply a $(n - k - 2)$-dimensional square matrix with a triangle in the upper-left. We can now use these findings to define an efficient algorithm that only has to keep the "shortcut" edges in memory, rather than *all* edges (Algorithm 10).

---

**Algorithm 10** Sample Newman–Watts–Strogatz model

---

1: **procedure** NEWMANWATTSSTROGATZEFFICIENT($n, k, p$)
2:   *RingLattice* $(n, k)$
3:   $T_i \leftarrow \left(\frac{k}{2} + 1\right)(n - k - 1)$ ▷ Start of bottom triangle
4:   $e_s \leftarrow Binomial\left(\frac{nk}{2}, p\right)$ ▷ Number of "shortcut" edges
5:   $inds \leftarrow SampleNoReplacement\left(e_s, 0, \frac{n(n-k-1)}{2} - 1\right)$
6:   **for all** $i \in inds$ **do**
7:     **if** $i < T_i$ **then** ▷ Top rectangle
8:       $u \leftarrow \left\lfloor \frac{i}{n-k-1} \right\rfloor$
9:       $c \leftarrow i \bmod (n - k - 1)$
10:    **else** ▷ Bottom triangle
11:      $t \leftarrow i - T_i$
12:      $r \leftarrow n - k - 3 - \left\lfloor \frac{\sqrt{-8t + 4(n-k-1)(n-k-2) - 7} - 1}{2} \right\rfloor$
13:      $u \leftarrow r + \frac{k}{2} + 1$
14:      $c \leftarrow t - \binom{n-k-1}{2} + \binom{n-k-1-r}{2}$
15:    $w \leftarrow u + c + \frac{k}{2} + 1$
16:    Output edge $(u, w)$
17:
18: ▷ Sample $n$ integers from *Uniform* $(a, b)$, no replacement
19: ▷ Algorithm attributed to Robert Floyd
20: **procedure** SAMPLENOREPLACEMENT($n, a, b$)
21:   $samples \leftarrow HashTable$ (*empty*)
22:   **for** $r \leftarrow b - n + 1$ to $b$ **do**
23:     $v \leftarrow Uniform(a, r)$
24:     **if** $v$ already exists in *samples* **then**
25:       Add $r$ to *samples*
26:     **else**
27:       Add $v$ to *samples*
28:   Return *samples*

---

## Benchmarking experiment

To benchmark network generation runtime and memory consumption, we used NetworkX, iGraph, and NGG to simulate 10 replicate networks of various sizes, and we used the GNU `time` command line tool to measure total runtime and peak memory usage. We chose to explore Complete, Erdős–Rényi, Barabási–Albert, and Newman–Watts–Strogatz graphs in



this benchmarking experiment due to their popularity in modeling social contact networks in epidemiological studies.

In addition to the number of nodes in the network ($n$), the Erdős–Rényi, Barabási–Albert, and Newman–Watts–Strogatz models have additional parameters that controls the expected degree ($E_d$) of the network; the choice of $E_d$ = 40 was made arbitrarily, and the same trend was observed for $E_d$ = 10 and $E_d$ = 20. All tools are single-threaded, and all runs were executed sequentially on an 8-core 2.0 GHz Intel Xeon CPU with 8 GB of memory.

The results of the benchmarking experiment can be found in Figure 1. iGraph was excluded from the Newman–Watts–Strogatz simulations because iGraph does not support sampling from the Newman–Watts–Strogatz model. Furthermore, NetworkX was unable to run to completion on larger network sizes due to memory requirements that exceeded the 8 GB memory of the benchmarking machine. In all scenarios, NGG was the fastest and least memory-intensive of the three tools.  With respect to Complete graphs, NGG is marginally faster than NetworkX and iGraph, and the peak memory usage of NGG is orders of magnitude smaller than both NetworkX and iGraph, with the gap widening as network size grows. With respect to Erdős–Rényi graphs, NGG is ~4× faster than NetworkX and ~1.5× faster than iGraph, and its peak memory usage is orders of magnitude smaller than both tools, with the gap again widening as network size grows. With respect to Barabási–Albert graphs, NGG is ~4× faster than NetworkX and ~1.5× faster than iGraph, and its peak memory usage is consistently ~20× smaller than NetworkX and ~3× smaller than iGraph. With respect to Newman–Watts–Strogatz graphs, NGG is ~3× faster than NetworkX, and its peak memory usage is ~100× smaller than NetworkX, with the gap widening as network size grows. Importantly, aside from the Barabási–Albert and Newman–Watts-Strogatz models, all network models implemented in NGG have constant memory usage regardless of network size.

## CONCLUSIONS

We introduce NiemaGraphGen (NGG), a memory-efficient graph generation tool that enables the simulation of global-scale contact networks. We benchmarked NGG against the two most popular network simulation tools, NetworkX and iGraph, and we showed that NGG was consistently fastest and had orders of magnitude lower memory consumption than the other tools (typically constant with respect to network size).

## AVAILABILITY OF SOURCE CODE AND REQUIREMENTS

- Project name: NiemaGraphGen (NGG)
- Project home page: https://github.com/niemasd/NiemaGraphGen
- Docker Hub page: https://hub.docker.com/r/niemasd/niemagraphgen
- Operating system(s): Platform independent
- Programming language: C++
- Other requirements: C++11 or higher
- License: GNU GPL v3.0
- RRID: SCR_021936

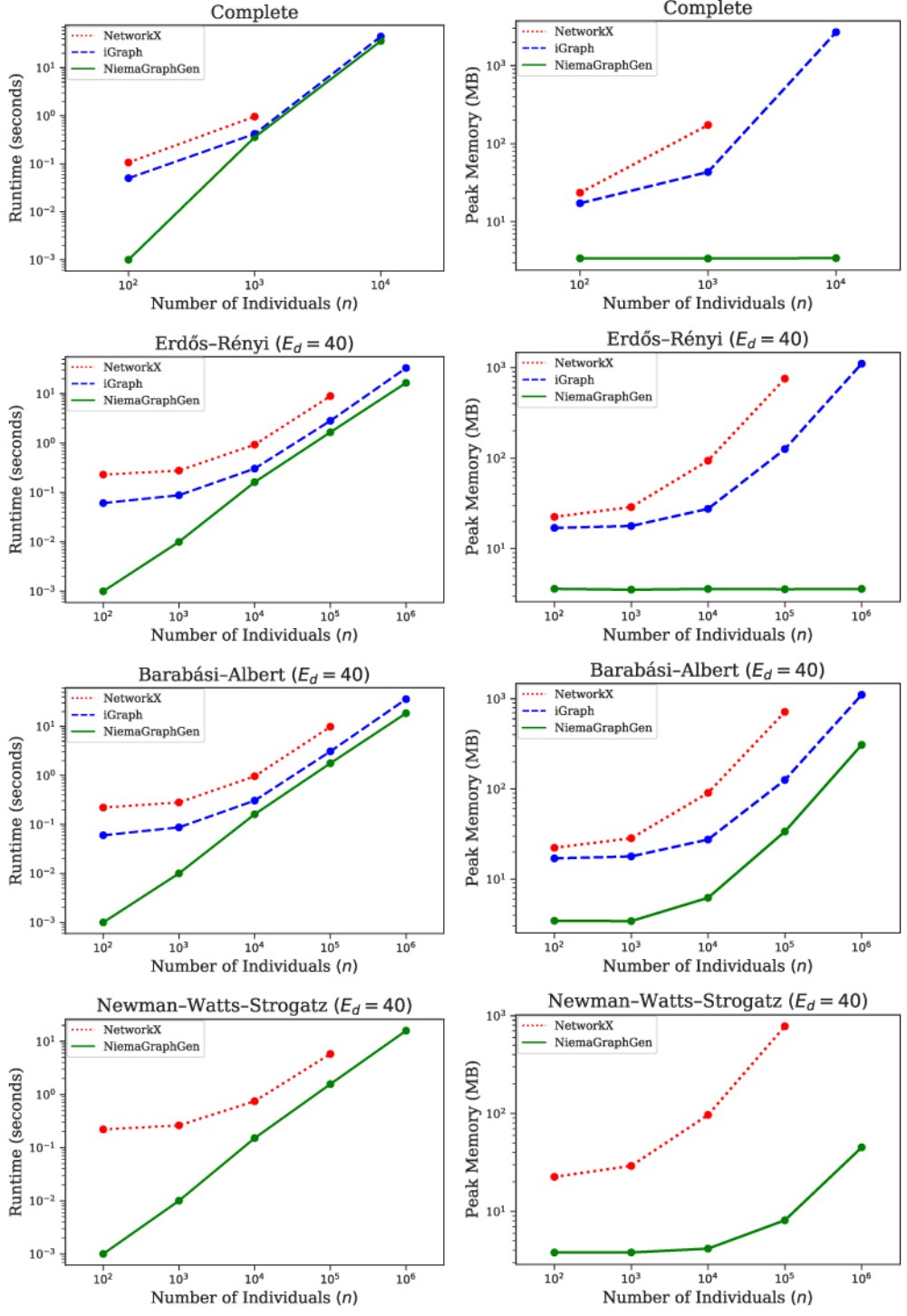

**Figure 1.** Benchmarking results. Total runtime (left) and peak memory usage (right) for NetworkX, iGraph, and NGG for various network models and sizes. Each point is the average of 10 replicates, and error bars (which are smaller than the marker sizes) represent 95% confidence intervals. All tools are single-threaded, and all runs were executed sequentially on an 8-core 2.0 GHz Intel Xeon CPU with 8 GB of memory.

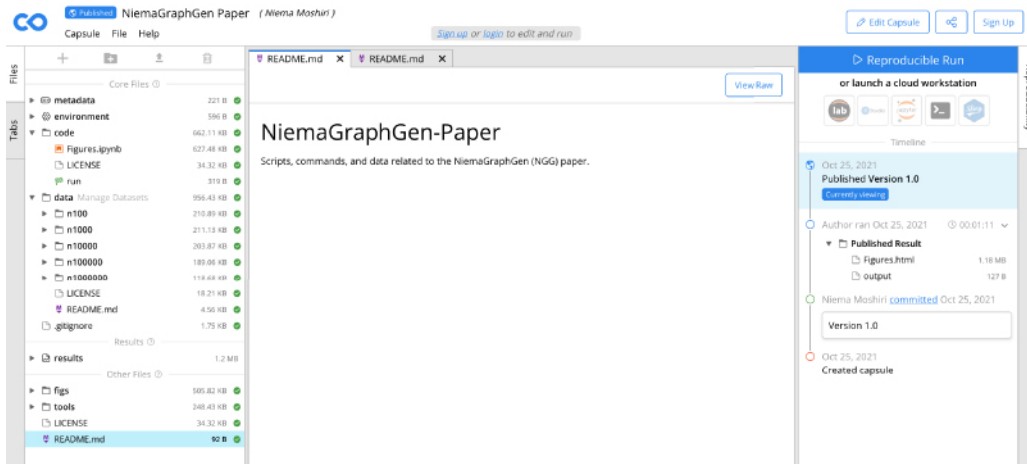

**Figure 2.** An executable Code Ocean compute capsule for NiemaGraphGen that can be launched on a cloud workstation. https://doi.org/10.24433/CO.4009211.v1

## DATA AVAILABILITY

The data sets supporting the results of this article, along with all relevant scripts and commands, are available in the following GitHub repository:

https://github.com/niemasd/NiemaGraphGen-Paper.

The same data and scripts can be found in the following portable Code Ocean environment (Figure 2) [19]. Snapshots of the code is also available in the *GigaScience* GigaDB repository [20].

## DECLARATIONS
## LIST OF ABBREVIATIONS

CPU: Central Processing Unit; GB: gigabyte; GPU: Graphics Processing Unit; NGG: NiemaGraphGen.

## ETHICAL APPROVAL

Not applicable.

## COMPETING INTERESTS

The authors declare that they have no competing interests.

## FUNDING

This work has been supported by the National Science Foundation (NSF) grant NSF-2028040 to N.M.

## AUTHOR'S CONTRIBUTIONS

N.M. implemented the software tool described in this manuscript, designed and executed the benchmarking experiment, and wrote the manuscript.

## ACKNOWLEDGEMENTS

We would like to thank Jonathan Pekar, Joel O. Wertheim, Michael Worobey, and Tajana Rosing for fruitful conversations.

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
