## [Reviewer Report]

Reviewer name and names of any other individual's who aided in reviewerFabian LorigDo you understand and agree to our policy of having open and named reviews, and having your review included with the published manuscript. (If no, please inform the editor that you cannot review this manuscript.)YesIs the language of sufficient quality?YesPlease add additional comments on language quality to clarify if neededIs there a clear statement of need explaining what problems the software is designed to solve and who the target audience is? YesAdditional CommentsIs the source code available, and has an appropriate Open Source Initiative license <a href="https://opensource.org/licenses" target="_blank">(https://opensource.org/licenses)</a> been assigned to the code?YesAdditional CommentsThe code is available via github and as docker container.As Open Source Software are there guidelines on how to contribute, report issues or seek support on the code?YesAdditional CommentsIs the code executable?YesAdditional CommentsIs installation/deployment sufficiently outlined in the paper and documentation, and does it proceed as outlined?YesAdditional CommentsIs the documentation provided clear and user friendly?YesAdditional CommentsIs there a clearly-stated list of dependencies, and is the core functionality of the software documented to a satisfactory level?YesAdditional CommentsHave any claims of performance been sufficiently tested and compared to other commonly-used packages? YesAdditional CommentsThe framework has been evaluated against NetworkX and iGraph, two libraries for generating graphs. However, these libraries do not explicitly claim that their purpose is the generation of large-scale graphs and other approaches/research in the field of large-scale graph generation is not discussed. Are there (ideally real world) examples demonstrating use of the software? NoAdditional CommentsThe authors outline the relevance of contact networks , for instance, in epidemic simulation experiments. However, the application of the software for this purpose is not demonstrated.Is automated testing used or are there manual steps described so that the functionality of the software can be verified?NoAdditional CommentsAny Additional Overall Comments to the AuthorThe outputs of the framework can be stored in the FAVITES tab-delimited edge list format as well as a (more efficient) binary output format. It should be outlined whether these formats are proprietary and how the import/export of graphs between NGG and other common libraries (e.g., NetworkX and iGraph) is possible.RecommendationMinor Revisions

---

## [Reviewer Report]

Reviewer name and names of any other individual's who aided in reviewerHans Zauner (editor)Do you understand and agree to our policy of having open and named reviews, and having your review included with the published manuscript. (If no, please inform the editor that you cannot review this manuscript.)YesIs the language of sufficient quality?YesPlease add additional comments on language quality to clarify if neededIs there a clear statement of need explaining what problems the software is designed to solve and who the target audience is? YesAdditional CommentsBut see the comments below.Is the source code available, and has an appropriate Open Source Initiative license <a href="https://opensource.org/licenses" target="_blank">(https://opensource.org/licenses)</a> been assigned to the code?YesAdditional CommentsAs Open Source Software are there guidelines on how to contribute, report issues or seek support on the code?YesAdditional CommentsIs the code executable?Unable to testAdditional CommentsIs installation/deployment sufficiently outlined in the paper and documentation, and does it proceed as outlined?Unable to testAdditional CommentsIs the documentation provided clear and user friendly?YesAdditional CommentsIs there a clearly-stated list of dependencies, and is the core functionality of the software documented to a satisfactory level?YesAdditional CommentsBut see comments below.Have any claims of performance been sufficiently tested and compared to other commonly-used packages? Not applicableAdditional Commentssee comments belowAre there (ideally real world) examples demonstrating use of the software? NoAdditional CommentsA real world example would greatly benefit the paper to demonstrate a use case for the tool.Additional CommentsAny Additional Overall Comments to the AuthorThe submission was previously reviewed for publication at GigaByte's sister journal, GigaScience, where I was the handling editor.  The overall impression during the previous review process was that the contribution is not substantial enough for the high standards of GigaScience. However, GigaByte also considers very short Technical Release articles and as such the submission is in scope for the journal, in principle.  Some of the remarks of the previous review process should still be addressed. For example, one previous reviewer pointed out that there are many very large scale network epidemic simulations ( e.g. by Madhav Marathe and Stephen Eubank), and tools are available to generate very large graphs (e.g. by Kalyan Perumalla). This work should be cited and briefly discussed, to put the tool into context.   I take from the previous review that , in today's context, NetworkX and iGraph may not be appropriate tools for massive network simulations. Also this objection should be at least discussed, and the use case for your tool better explained in light of this. I recommend to do this with a "worked" example (test data can be hosted in our repository GigaDB).  Another point of the previous review, that should also be addressed for publication in GigaByte, was the recommendation to include standard deviations for the graphs and benchmark whether results are within a desired confidence interval.RecommendationMinor Revisions